# Identifying Three-Dimensional Facial Fluctuating Asymmetry in Normal Pediatric Individuals: A Panel Assessment Outcome Study of Clinicians and Observers

**DOI:** 10.3390/jcm8050648

**Published:** 2019-05-10

**Authors:** Pang-Yun Chou, Rafael Denadai, Shih-Heng Chen, Hsiao-Jung Tseng, Chih-Kai Hsu, Sheng-Wei Wang, Rami Hallac, Chih-Hao Chen, Alex A. Kane, Lun-Jou Lo

**Affiliations:** 1Department of Plastic and Reconstructive Surgery and Craniofacial Research Center, Chang Gung Memorial Hospital, Chang Gung University, Taoyuan 333, Taiwan; chou.asapulu@gmail.com (P.-Y.C.); denadai.rafael@hotmail.com (R.D.); shihheng@icloud.com (S.-H.C.); kkhsu0315@gmail.com (C.-K.H.); chchen5027@gmail.com (C.-H.C.); 2Clinical Trial Center, Chang Gung Memorial Hospital, Taoyuan 333, Taiwan; allebjht@gmail.com; 3Department of Biomedical Engineering, National Yang-Ming University, Taipei 112, Taiwan; r025876@gmail.com; 4Analytical Imaging and Modeling Center, Children’s Medical Center, Dallas, TX 75207, USA; rami.hallac@childrens.com (R.H.); alex.kane@utsouthwestern.edu (A.A.K.); 5Department of Plastic Surgery, UT Southwestern, Dallas, TX 75390, USA

**Keywords:** normal children, facial asymmetry, clinicians, observers, panel assessment, three-dimensional images

## Abstract

This study measured three-dimensional facial fluctuating asymmetry in 600 normal and healthy Taiwanese individuals (6 to 12 years old) and assessed the perceptions of increasing levels of facial fluctuating asymmetric severity by using a panel composed of 20 clinicians (surgical professionals), as well as 20 adult and 40 pre-adolescent observers. On average, this normal cohort presented a facial fluctuating asymmetry of 0.96 ± 0.52 mm, with 0.52 ± 0.05, 0.67 ± 0.09, 1.01 ± 0.10, and 1.71 ± 0.36 mm for levels I, II, III, and IV of severity, respectively. For all categories of raters, significant differences in the average symmetry–asymmetry scale values were observed, with level I < level II < level III = level IV (all *p* < 0.01, except for level III vs. IV comparisons with *p* > 0.05). For level I, pre-adolescent observers presented a significantly (*p* < 0.05) higher symmetry–asymmetry scale value than adult observers, with no significant (all *p* > 0.05) differences for other comparisons. For overall facial asymmetry and levels II, III, and IV, no significant (all *p* > 0.05) differences were observed. This study reveals that the normal pediatric face is asymmetric and the panel assessment of facial fluctuating asymmetry was influenced by the level of severity and the category of raters and contributes to the literature by revealing that pre-adolescent raters present a similar or higher perception of facial asymmetry than adult raters.

## 1. Introduction

From an ecology and evolutionary biology standpoint, perfect bilateral symmetry is defined as the optimal outcome of the development of bilateral traits in the absence of perturbations [1]. For human facial anthropometrics, perfect bilateral symmetry almost never exists, because random variations in asymmetry, within limits, have been recognized as normal and are called facial fluctuating asymmetry [2,3,4,5,6,7,8,9,10,11,12,13,14,15]. When facial asymmetry is clinically obvious (called facial asymmetric deformity, which is more commonly directional rather than fluctuating), surgical or nonsurgical treatment may be required [3,4,5]. Since the restoration of symmetry is the main goal of facial asymmetric deformity reconstruction, establishing facial fluctuating asymmetry by specific ethnic and age groups is crucial as it may be adopted as a target of treatment as well as applied as normative data for genetics, orthodontics, and surgical disciplines [6,7,8,9,10,11].

The recent introduction of three-dimensional (3D) surface image technology revolutionized facial asymmetry-based investigations by allowing us to acquire consistently high-quality 3D facial imaging without exposure to radiation [7,8,9,10,11,12,13,14,15]. For human faces, which are complex structures, these 3D image capturing systems offer a number of distinct advantages, including minimal invasiveness; quick capture speed; the ability to archive images for subsequent analyses; high-quality images without distortion of spatial form, shape, and size, resulting in a lifelike rendering; a realistic facial view with images moving from one side to the other; and a high degree of reproducibility, reliability, and validity [12,13,14,15].

Despite the advantages of the computer-based measurement of facial asymmetry [7,8,9,10,11,12,13,14,15], one of the most common outcome measures adopted in facial asymmetry studies is panel assessment, a metric based on perceptions from different categories of raters, including clinicians (professionals using medical- or dental-based judgments) and observers (laypersons with no formal specialized training) [5,8,12,16,17,18,19,20,21,22]. These panel assessment-based studies have been widely adopted to identify the presence of facial asymmetry or to quantify facial asymmetry, with relevant clinical applications, for example, for outcome studies comparing various therapeutic modalities because clinicians represent the specialized judgment of pre- and post-treatment images, whereas observers symbolize the external validity in terms of the public members’ observations of therapeutic effects [5,8,12,16,17,18,19,20,21,22].

In this setting, although a growing number of studies have used panel assessment of 3D facial images [8,12,20,21,22], no 3D image-based clinical investigation has formally focused on the assessment of facial asymmetry in a normal sample of elementary school individuals (aged between 6 and 12) based on a panel composed of their peers, namely pre-adolescent observers (aged 12). Moreover, when assessing 3D facial fluctuating asymmetry, comparisons between clinicians and pre-adolescent observers have not been addressed to date. When evaluating pediatric patients with facial deformities the particular perception of pre-adolescent observers is clinically relevant because it acts as a simulation of face-to-face social interactions between patients and their peers, corresponding to their daily life at school. Considering that facial asymmetry has been regarded as a continuum, ranging from facial fluctuating asymmetry in the normal population to patients with a myriad of facial asymmetric conditions, reliable information regarding the panel assessment of facial asymmetry in the normal population needs to be established first of all.

The purposes of this study were to measure the 3D facial fluctuating asymmetry in a cohort of normal pediatric individuals and assess the various severity levels of facial fluctuating asymmetry using a panel composed of surgical professionals and adults, as well as pre-adolescent observers. Based on various degrees of background knowledge and experience [5,8,12,16,17,18,19,20,21,22], we hypothesized that surgical professionals versus pre-adolescent observer comparisons present significant differences when identifying facial symmetry or asymmetry in normal individuals, distributed in four levels of facial fluctuating asymmetric severity.

## 2. Materials and Methods

This was an institutional review board-approved cross-sectional study involving the collection of 3D facial images for panel assessment. Normal, healthy pediatric individuals, aged between 6 and 12 years and with no known craniofacial abnormalities, were randomly recruited on a voluntary basis from elementary schools in Taiwan. Each potential participant was clinically screened extra- and intra-orally by a multidisciplinary craniofacial team at the Chang Gung Craniofacial Center by using itemized criteria for the etiology of facial asymmetric deformity, including congenital anomaly, developmental condition, and acquired injury or disease [3]. Individuals were excluded from the study if they had (1) mixed or uncertain ethnicity (non-Taiwanese); (2) presumed or confirmed diagnosis of any syndromic or nonsyndromic craniofacial deformity; (3) a history of facial trauma; or (4) undergone orthodontic treatment or facial surgery. All parents of the included individuals provided written consent for participation in this study.

### 2.1. 3D Facial Image Acquisition

3D facial photographs (digital stereophotogrammetric surface images) of all included individuals were acquired using the 3dMD system (3dMD LLC, Atlanta, GA, USA) under the following standard conditions: A permanent installation with fixed ambient lighting and system and fixed individual positioning, including individuals with a natural head position, relaxed facial musculature, a closed mouth, and thin elastic nylon caps to keep the hair away from the face [7,9]. The system was calibrated before every capture process. An informatic-trained professional, a member of the Chang Gung Craniofacial Research Center, was responsible for the whole 3D image acquisition process.

### 2.2. 3D Data Processing

All 3D datasets were processed and analyzed using the 3dMD Vultus software package (version 2.2, 3dMD Inc., Atlanta, GA, USA) and MATLAB, based on a previously described and validated method [9]. Initially, a perfectly symmetric model, with known left and right point correspondences, was created from a 3D image using customized programs formed in MATLAB (MATrix LABoratory) and was used as a reference template to calculate the facial asymmetry in each individual’s 3D image. Thirty-two anatomical landmarks (Figure 1) were identified on this template, using each individual’s 3D texture map image (3dMD Vultus software). For each individual’s 3D image, registration was performed through rigid translation and rotation to match the template using the 32 landmarks. An anisotropic scaling (length, width, and height) of the template was also performed to match each individual’s 3D image. The template was subsequently deformed to each individual’s 3D image using a thin-plate spline algorithm and closest-point deformation. For this stage, 40 landmarks were constructed digitally and dispersed into four layers surrounding the skull apex, which is the point where the y-axis (vertical dimension) meets the skull (Figure 1). The layers were defined by descending 10 degrees from the apex, one after another, and each of them consisted of 10 evenly distributed landmarks. A polygonal indexing of all the 3D images matched that of the template, allowing for the creation of a mean (composite) head from the deformed templates using point-wise average. A composite head was created by combining the 3D deformed templates of all the healthy normal pediatric individuals included (Figure 2).

For facial asymmetry measurement, the origin of the coordinate system for each individual’s 3D model was calculated using the nearest projection of the nose tip point to the line connecting bilateral tragus points. The Euclidean distances from the origin of the coordinates to each anatomical landmark point on the deformed template were then automatically measured. The facial asymmetry for each individual was represented by the absolute difference between the left and right corresponding points. The facial area was defined by counting the points that had shorter Euclidean distances to nose tip than those between the nose tip and gnathion. In total, 4912 pairs per model were calculated in the face area. The mean asymmetry was calculated for all points.

### 2.3. Levels of Facial Fluctuating Asymmetry

All included individuals were linearly distributed based on facial asymmetry incremental values. Facial fluctuating asymmetry was then stratified in four levels of severity (levels I to IV) from the minimum to maximum values, with 150 individuals per level of severity.

### 2.4. Stimuli Processing

Twenty of the 600 individuals were selected to be rated by the clinicians and observers. To this end, five individuals were selected from each level of facial asymmetry. A random number generator was used to avoid bias when picking the 20 individuals. To prepare the 3D facial image set for panel assessment, static (full face frontal view) and dynamic photographic views were obtained for each individual. A standard reference frame (horizontal, coronal, and sagittal planes) was established for all these 3D photographs before capturing. For 3D moving images, an animation rotating from the frontal view to the right and left profile views (Appendix A) was captured using the 3dMD Vultus software package. The static and dynamic images (distributed on the right and left side of each slide, respectively) were presented in a fixed random sequence in a timed Microsoft PowerPoint presentation (Microsoft Corporation, Redmond, WA, USA) on a 15-inch MacBook Pro (MacBook Pro, Apple, Inc., Cupertino, CA, USA). Each colored slide was displayed for 6 s, with a 2 s break between each of the slides.

### 2.5. Panel Assessment

The 3D image set was rated, by a panel composed of three categories of raters, using a binomial facial symmetry versus asymmetry grading system. For the observer-based assessment, two groups of laypersons with no specialized professional training (dental, medical, and psychologist background) were recruited as follows: A total of 40 pre-adolescent children (20 females, all aged 12 years) randomly recruited from 6th-grade students who were not participating in the 3D image data collection and were from elementary schools other than those of the included individuals and 20 adults (10 women, aged 19–44) randomly recruited from members of the general public at public places, based on incidental contacts. For the clinician-based assessment, 20 surgical professionals (10 males; 4 to 15 years after board certification in plastic surgery) were randomly selected from members of the Taiwan Society of Plastic Surgery and Asiatic plastic surgeons visiting the Chang Gung Craniofacial Center. Using one spreadsheet per slide, the rater wrote down his/her perception of facial symmetry versus asymmetry that each individual under appraisal was believed to present. For this, all raters received the same instructions before appraisal of the 3D image set, as follows: (1) A description of the purpose of the study (how do you perceive the face of this individual?); (2) a definition of the facial symmetry versus asymmetry parameter (one side of the face may be similar or different to the other side) was provided, with each rater paraphrasing it with no comprehension problems; and (3) an explanation of how to fill out the spreadsheet (mark the circle corresponding to the binominal choice, facial symmetry or facial asymmetry options). All raters had no relationship to the included individuals, were masked to the facial fluctuating asymmetry level status of each image, and were not allowed to go back in the presentation. No compensation was provided for their participation.

### 2.6. Statistical Analysis

In the descriptive analysis, the mean was used for metric variables and percentages were given for categorical variables. The data distribution was verified through the Kolmogorov‒Smirnov test and the Kruskal–Wallis test, chi-squared test, one-way repeated-measures ANOVA, Cronbach’s alpha test, and Fleiss’s kappa were adopted for statistical comparisons. Two-sided values of *p* < 0.05 were considered statistically significant. All analyses were performed using SPSS version 22.0 (Chicago, IL, USA).

## 3. Results

The 3D facial images of 600 normal and healthy Taiwanese pediatric individuals with no known craniofacial abnormality were analyzed. The male-to-female ratio was 1 and the mean age was 7.98 ± 1.19 years (Table 1).

### 3.1. Levels of Facial Fluctuating Asymmetry

On average, this cohort presented a facial fluctuating asymmetry of 0.96 ± 0.52 mm, with 0.52 ± 0.05, 0.67 ± 0.09, 1.01 ± 0.10, and 1.71 ± 0.36 mm for levels I, II, III, and IV, respectively (Table 1, Figure 3). No significant (all *p* > 0.05) differences were observed between males and females for average facial asymmetry in all levels. No significant (all *p* > 0.05) differences were noted between the individuals (*n* = 20) per level of facial fluctuating asymmetry adopted for the panel assessment and the total level groups (*n* = 600) for the mean age and facial asymmetry values.

### 3.2. Panel Assessment

For all categories of raters, significant differences for the average symmetry–asymmetry scale values were observed between the levels of facial fluctuating asymmetry, with level I < level II < level III = level IV (all *p* < 0.01, except for level III versus level IV comparisons with *p* > 0.05) (Table 2). When the facial fluctuating asymmetry increased from level I to IV, the number of raters perceiving a higher number of asymmetric individuals within each level exhibited a significant (all *p* < 0.01) trend in linear and quadratic changes.

For facial asymmetry overall, no significant (*p* > 0.05) differences were noted for all comparisons between categories of raters. For level I of facial fluctuating asymmetry, pre-adolescent observers presented a significantly (*p* < 0.01) higher symmetry–asymmetry scale value than adult observers, with no significant differences for pre-adolescent observers versus plastic surgeons (*p* = 0.08) and adult observers versus plastic surgeons (*p* = 0.99). For levels II, III, and IV of facial fluctuating asymmetry, no significant (all *p* < 0.05) differences were observed for comparisons between plastic surgeons, pre-adolescent observers, and adult observers (Table 2).

Each category of rater demonstrated acceptable internal consistency (Cronbach’s alpha coefficient = 0.681 to 0.864), with no significant (all *p* > 0.05) differences between the raters’ categories for the distribution of ratings per level of facial asymmetry. Interrater reliability was considered as slight agreement (*k* = 0.070 to 0.136) (Table 3).

## 4. Discussion

Conceptually, facial symmetry relates to a perfect correspondence in the size, shape, and arrangement of the anatomical features on both sides of the midsagittal plane [1]. However, human faces are structurally left–right symmetrical content-wise, but not size-wise or function-wise. Therefore, asymmetric faces would be, to some extent, the norm [2,3,4,5,6,7,8,9,10,11]. In a recent and well-delineated 3D facial image study, Kane’s group found that 99.7% of healthy pediatric individuals aged between 0 and 17.8 had less than 3.2 mm of facial asymmetry [9]. Although they described no significant difference in facial asymmetry between ethnicities, only 4% Asian pediatric individuals were included, for a total sample of 533 African American, Asian, Hispanic, and Caucasian pediatric individuals [9]. The levels of severity for facial fluctuating asymmetry or perceptions of facial asymmetry by panel assessment were not provided because it was beyond the scope of the work [9].

By adopting the same 3D facial image methodological design described by Kane’s group [9], all Taiwanese pediatric individuals with no craniofacial deformity, trauma, or operations could be fitted within the previously described cutoff of 3.2 mm [9], with all included individuals presenting facial asymmetry values between 0.315 and 2.602 mm. We also defined four levels of facial asymmetric severity, ranging, on average, from 0.52 to 1.71 mm for levels I and IV, respectively. These values can be adopted in outcome-based research to measure the pre- and postoperative status of patients with craniofacial deformities and aged between 6 and 12 years old. It may also be valuable for the improvement of planning and decision-making processes of multidisciplinary teams by determining the worth of pursuing conservative or surgical management of mild facial asymmetric deformity or residual facial asymmetry after surgical management [3,4,23,24,25]. It may be further applied for the preoperative and postoperative counseling of patients and parents regarding the acceptance or declination of correction of facial asymmetry when using normal-based data as the parameter of discussion.

Based on these normal individual-based facial asymmetry data, we assessed the perceptions of various levels of 3D facial fluctuating asymmetric severity by using a panel analysis, composed of plastic surgeons, pre-adolescent student lay observers, and adult lay observers. For this analysis, colored 3D frontal view and 3D full face animations rotating from the frontal view to the right and left profile views were adopted because it allowed the raters to perceive temporally the 3D nature of the facial images instead of only static greyscale 3D images or two-dimensional (2D) projections of frontal and/or profile views. Since no similar investigation existed with the same study design, the previous findings may not be directly comparable. Most previous 3D facial image studies have included mixed samples (normal individuals with a wide age range and patients with craniofacial deformities), panel compositions (no definition of the raters background, adult lay observers, and clinicians, such as dental care professionals, general dental professionals, orthodontists, and surgeons), types of image set (only static frontal views, manipulated or simulated facial images, greyscale images, and colored images), and assessment methods (from midline deviations to facial asymmetry overall perceptions) [8,20,21,22].

Contrary to our initial hypothesis, plastic surgeons exhibited no significant differences for perceptions of facial symmetry–asymmetry in all levels of severity, compared with adults and pre-adolescent observers. These findings revealed that various levels of facial fluctuating asymmetry were also identified or perceived by lay observers, who had no expertise (specialized training) in evaluating facial asymmetries. In the literature, there are conflicting results regarding the differences between clinicians (surgical and orthodontic professionals) and lay observers. Clinicians are considered more familiar with the evaluation and treatment of facial asymmetry, presenting higher or equal values for 2D and 3D symmetry–asymmetry identification compared with lay observers [16,17,20,21,22,26,27].

Particularly in 3D photography studies, as previously described for lay observers and clinicians (surgical and orthodontic professionals) [21], we revealed that an increase in 3D facial asymmetric severity was rated, on average, as more asymmetric by all three categories of raters. These results were significant for the increase of facial asymmetry from levels I to III of severity, displaying no significant differences for levels III versus IV comparisons in all categories of raters. We hypothesized that facial symmetry/asymmetry-related values between levels III and IV may be the plateau of asymmetry perception or identification in normal individuals, with these values potentially acting as a parameter to distinguish normal individuals from abnormal patients. However, although our normal pediatric sample-based data provide an indication of where the limit between normal and abnormal facial asymmetric values may be, it is not sufficient to establish the exact threshold that separates facial fluctuating asymmetry and facial asymmetric deformity. Even though it could be straightforward to numerically distinguish normal individuals from patients with severe facial asymmetric deformities, this is not the case for mild deformities. Patients with mild facial asymmetric deformities, such as those with cleft lip nose deformity, mild craniofacial microsomia (soft-tissue deficiency type I and Pruzansky–Kaban type I mandibular hypoplasia), or mild Parry–Romberg syndrome (cutaneous atrophy only) [13,23,24,25,28], may have 3D numerical values in millimeters similar to normal individuals who are stratified within levels III and IV of facial fluctuating asymmetric severity. For future studies, it is necessary to compare our current data to those of patients with facial asymmetric deformities who present mild to severe levels of severity. In addition, the different levels of facial fluctuating asymmetric severity could be further stratified in relation to the treatment needs (normal, socially acceptable, or deemed appropriate for correction), which may help to establish the numerical target of facial asymmetry reconstructions.

Our findings demonstrate that pre-adolescent observers may judge normal individuals as having more (level I) or equivalent (levels II to IV) facial asymmetry compared with adult raters. From the psychological literature perspective, various aspects related to facial symmetry have been investigated using perceptions from pediatric and adult individuals, with most of the existing evidence focused on the relation between facial symmetry–asymmetry and facial attractiveness [29,30,31,32,33,34]. The specific influence of symmetry on the attractiveness judgment typically emerges and matures after five and nine years of age [32,35]. However, asymmetry appears to be not only a significant determinant of perception of facial attractiveness but also to have an effect on social desirability, as well as the development and maintenance of interpersonal relationships [36,37]. Facial differences may affect an individual’s overall psychological functioning, as they may contribute to challenges with self-image, self-esteem, and emotional development [38]. The literature confirms that having physically distinct faces often leads to more situations of psychological struggle and bullying [39,40]. As a consequence of the extreme importance of peer approval during the developmental stage of childhood for humans building their self-concept and social identity [41,42,43], children with facial differences may experience a higher degree of teasing and bullying related to their facial appearance [39,40,44,45]. Given the increased importance of social experiences with peers during school age, the negative association between facial asymmetry, social perceptions, psychosocial functioning, and the facial asymmetry as perceived by pre-adolescent observers in normal individuals, it can be argued that the child population that is the target of our investigation may be at an increased risk with regard to social difficulties and psychosocial functioning.

From a practical perspective, although clinicians usually make technical recommendations regarding the course of therapy (e.g., they may recommend or argue against the initial or revision surgery) to patients with facial differences (including asymmetry) and their parents, the psychosocial well-being of pediatric individuals with facial differences is intensely influenced by their social environment (primarily during the elementary school years), comprising the majority of face-to-face interactions and situations with laypersons (classmate peers) [39,40,41,42,43,44,45]. As described above, layperson perceptions of facial asymmetry may result in difficult situations (e.g., bullying through repeated harmful physical, verbal, social, and electronic media acts), thus determining how successful the psychosocial adjustment of children with facial differences will be. Interpreting the results of judgments by adults and pre-adolescent observers, both the disparity (level I of severity) and similarity (levels II and III) findings may be a source of psychosocial disturbance in a school environment and dissatisfaction with facial asymmetry treatment. For disparity perceptions in level I, the peers of pediatric individuals with facial asymmetry level I may judge their faces differently than school teachers, which may create problems, such as the delayed identification of a potential conflict. Although the similarity in levels II–IV does not obviate the possibility of difficult situations between pediatric individuals with facial differences and their classmate peers, it offers an opportunity to define preventive measures. Adults may identify children with facial differences and provide psychological support for them. They may also use educational measures to assist their peers in creating a supportive peer network and reinforce positive behaviors toward facial differences as the normality. These practical extrapolations from our data should be the target of future investigations using alternative methodological designs. Moreover, based on our current findings, it appears that adult panels (professionals and lay observers) and peer panels (pediatric lay observers) are equally relevant for future facial asymmetry-based research in the pediatric population. The decision on the type of panel assessment composition might depend on the research question under investigation or practical limitations, such as the school environment of pediatric individuals with facial differences.

Limitations in the current study should be noted and addressed by future research. Our data are restricted to individuals aged between 6 and 12, but we intentionally selected this age range because of school environment-related aspects. This age range was also based on a growing number of studies describing the surgical management of the facial asymmetry in patients within this age span [25,28,46,47,48,49,50,51,52]. As bone deformity has not been definitively proven to require surgical reconstruction in growing patients with facial contour asymmetry and no functional issues [25,28,46,47,48,49,50,51,52], 3D facial photography has emerged as a key tool to assess the magnitude of deformity and support surgical planning preoperatively, as well as to measure facial changes postoperatively, with no need for radiation exposure. Although our findings are limited to a relatively large total sample of 3D facial images from healthy and normal Taiwanese individuals, other groups are stimulated to collect facial fluctuating asymmetry values and their specific ethnic populations because it may generate a 3D facial image database worldwide. We did not distribute the facial asymmetry, by side or anatomical region, because our objective was to appraise the facial asymmetry overall. Although a global assessment of asymmetry was performed here, the distribution of local and body feature-specific asymmetries (e.g., nose versus chin asymmetries) may have affected the perception of overall asymmetry [21], deserving further evaluation. We adopted a binomial facial symmetry–asymmetry score. Other studies may test additional parameters (perfectly mirrored versus obvious left‒right mismatch) and Likert-type or visual analog scales in pre-adolescent observers. Similar to other groups presenting poor interrater reliability values for observers assessing 3D facial asymmetry in normal individuals [8], our interrater reliability was considered to show slight agreement, revealing provable variations in the perception between members of each category of raters. Other categories of professionals (psychologists, orthodontics, oral and maxillofacial surgeons, and otorhinolaryngologists, among others) and lay observers (pediatric individuals in childhood and adolescence phases, elementary school teachers, and patients with facial asymmetric deformities and their parents) should be included in future studies, because their opinions are necessary in relation to educational and treatment needs, resource allocation, and informed consent.

Despite these limitations, our current findings provide a benchmark for future medical research with extensive applications in clinical practice including diagnostic definitions, treatment planning, and postoperative evaluations of therapeutic outcomes and longitudinal follow-up (progression of untreated and treated pediatric individuals with facial asymmetry).

## 5. Conclusions

This study revealed that normal the Taiwanese pediatric facial form is asymmetric, the panel assessment of facial fluctuating asymmetry was influenced by the level of severity and the category of raters, and contributed to the literature by revealing that pre-adolescent observers present a similar or higher perception of facial asymmetry compared with adult clinicians and observers.

## Figures and Tables

**Figure 1 jcm-08-00648-f001:**
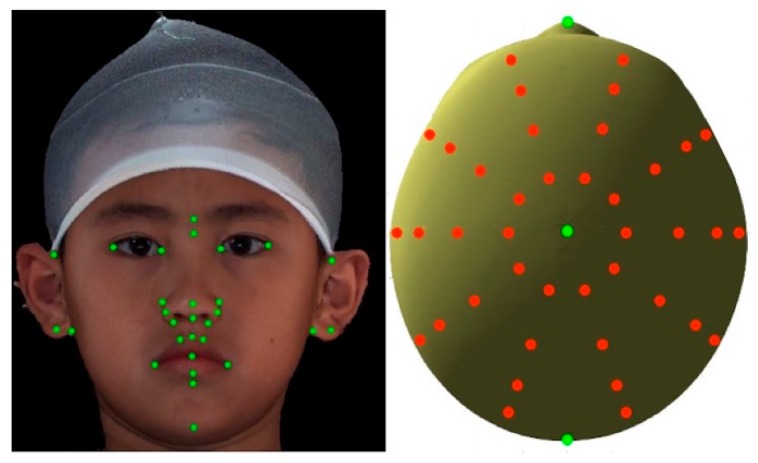
Three-dimensional images displaying (left) 32 manually-placed landmarks (green) on recognizable anatomy structures and (right) additional 40 digitally constructed landmarks (red) from top view.

**Figure 2 jcm-08-00648-f002:**
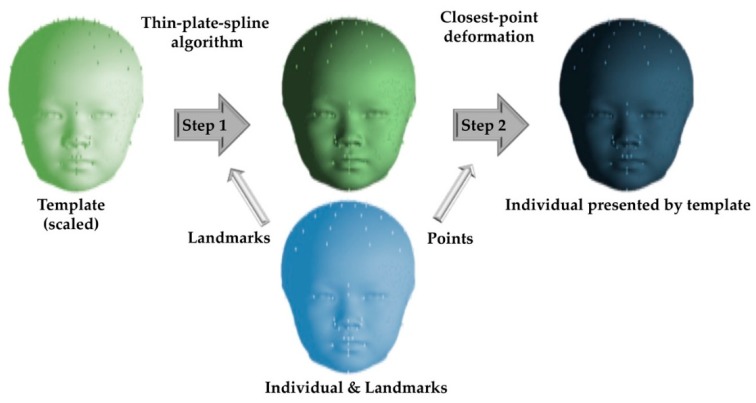
For facial asymmetry measurement, thin-plate splines and closest-point deformation were performed to deform a perfectly symmetric model to each individual’s three-dimensional image and establish point correspondence between the left and right sides of each individual. Facial asymmetry was calculated for a densely sampled set of anatomical landmark points for each healthy normal pediatric individual included.

**Figure 3 jcm-08-00648-f003:**
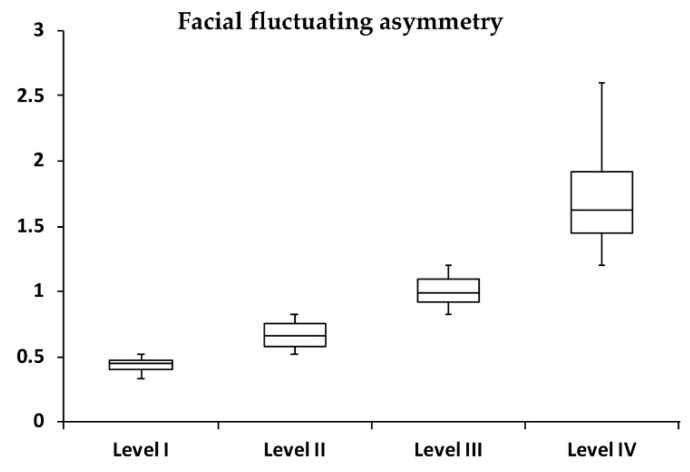
Box plot demonstrating the distribution of facial fluctuating asymmetry (*n* = 150 for level of asymmetry).

**Table 1 jcm-08-00648-t001:** Characteristics and levels of facial fluctuating asymmetric severity in normal pediatric individuals.

Parameters	Individuals (*n* = 600)
**Facial fluctuating asymmetry overall** m ± sd	0.96 ± 0.52 (0.327–2.602)
Male/Female *n* (%)	300 (50)/300 (50)
Age (years) m ± sd	7.98 ± 1.19
6–8 *n* (%)	432 (72)
9–12 *n* (%)	168 (28)
**Facial fluctuating asymmetric level I** m ± sd	0.44 ± 0.05 (0.327–0.518)
Male/Female *n* (%)	78 (52)/72 (48) *
Age (years) m ± sd	7.81 ± 1.16 *
6–8 years *n* (%)	115 (76.67)
9–12 years *n* (%)	35 (23.33)
**Facial fluctuating asymmetric level II** m ± sd	0.67 ± 0.09 (0.519–0.827)
Male/Female *n* (%)	70 (46.67)/80 (53.33) *
Age (years) m ± sd	7.99 ± 1.14 *
6–8 years *n* (%)	115 (76.67)
9–12 years *n* (%)	35 (23.33)
**Facial fluctuating asymmetric level IIII** m ± sd	1.01 ± 0.10 (0.830–1.119)
Male/Female *n* (%)	75 (50)/75 (50) *
Age (years) m ± sd	8.11 ± 1.32 *
6–8 years *n* (%)	103 (68.67)
9–12 years *n* (%)	47 (31.33)
**Facial fluctuating asymmetric level IV** m ± sd	1.71 ± 0.36 (1.204–2.602)
Male/Female *n* (%)	78 (52)/72 (48) *
Age (years) m ± sd	7.99 ± 1.12 *
6–8 years *n* (%)	99 (66)
9–12 years *n* (%)	51 (34)

*n*, number of individuals; m, mean; sd, standard deviation; *, *p* > 0.05 for all comparative analyses between all levels.

**Table 2 jcm-08-00648-t002:** Panel assessment of various levels of facial fluctuating asymmetry using the facial symmetry/asymmetry grading system.

Facial Fluctuating Asymmetry	Categories of Raters			*p*-Value **^,#^
	Plastic Surgeons	Adult Observers	Pre-Adolescent Observers	
Overall	2.60 ± 0.94	2.05 ± 1.10	2.74 ± 0.84	>0.05
Level I	1.40 ± 1.14	1.26 ± 1.15	2.17 ± 1.17	<0.05 ^‡^
Level II	2.65 ± 1.23	1.89 ± 1.15	2.38 ± 1.15	>0.05
Level III	3.20 ± 1.06	2.53 ± 1.54	3.26 ± 1.23	>0.05
Level IV	3.15 ± 1.35	2.53 ± 1.35	3.17 ± 1.15	>0.05
*p*-value *^,#^	<0.05 ^†^	<0.05 ^†^	<0.05 ^†^	–

Data presented as mean ± standard deviation based on the facial symmetry/asymmetry grading system (range, 1 to 4); –, not applicable; *, intracategory comparative analysis; **, intercategories comparative analysis; ^†^, level I < level II < level III = level IV (*p* < 0.01 for all comparisons, except for level III versus level IV comparisons with *p* > 0.05); ^‡^, *p* < 0.01 for comparison of pre-adolescent observers versus adult observers, with *p* > 0.05 for comparison of pre-adolescent observers versus plastic surgeons and adult observers versus plastic surgeons; ***^#^*** adjustment for multiple comparisons (Bonferroni corrections).

**Table 3 jcm-08-00648-t003:** Intra and interrater reliability for facial asymmetry/symmetry assessments.

Categories of Raters	Intra-Rater Reliability (Cronbach’s Alpha Coefficient)	Inter-Rater Reliability (*k*)
Plastic surgeons	0.792	0.136
Adult observers	0.864	0.105
Pre-adolescent observers	0.681	0.070

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
