# Peer review of "Identifying Three-Dimensional Facial Fluctuating Asymmetry in Normal Pediatric Individuals: A Panel Assessment Outcome Study of Clinicians and Observers"

_jcm, 2019, doi:10.3390/jcm8050648_

Round 1
Reviewer 1 Report
First of all, I would like to thank you all for the contribution you have made to the field of craniofacial anatomy. I believe that far too little research effort is placed into understanding normal variation in craniofacial development, and your article helps fill that void. My suggestions for improvement are as follows:
Introduction
-In the first paragraph you talk about fluctuating facial asymmetry and clinically relevant facial asymmetry. I think that it would be useful to distinguish between fluctuating asymmetry and facial asymmetry requiring clinical treatment, which is more commonly directional rather than fluctuating.
Methods
-On lines 97-100 you list the criteria for exclusion, but it was not clear if mixed or uncertain ethnicity needed to be present with craniofacial deformities or if these were two separate exclusion criteria. Logically I would assume that it’s the latter, but if so it would be better to rework this sentence to make it more clear.
-The facial masks in Figure 2 appear to be squished, such that the height is too small for the width of each image
-In section 2.3 you note that you divided the sample into even groups of 150 based on the severity of asymmetry. While I understand the logic behind this decision based on your panel assessment method, These groups are not inherently meaningful. It would be interesting to know what the distribution of fluctuating asymmetry is in this population. Even a simple dot plot for the entire sample or for the age groups in the sample in your Results section would be very informative.
Results
-The first paragraph in this section lists the age distribution for the sample. This may be a matter of personal preference, but I think this would be more clearly displayed as a table with the actual number of individuals in each age group listed with fluctuating asymmetry data.
-Similarly, section 3.1 lists the fluctuating asymmetry data for each of the severity groups. A box and whisker plot would be a useful way to visualize these data.
Conclusions
-I would suggest eliminating this section and placing this sentence at the beginning of the Discussion section, unless this section is required by the journal.
Thank you again for the opportunity to review this paper. I enjoyed reading it and I think that other craniofacial researchers will find it illuminating.
Author Response
May 8, 2019
Prof. Dr. Emmanuel Andrès
Editor-in-Chief,Journal of Clinical Medicine
Dear Prof. Dr. Andrès,
We would like to thank you and your reviewers for considering our manuscript entitled “Identifying Three-Dimensional Facial Fluctuating Asymmetry in Normal Pediatric Individuals: A Panel Assessment Outcome Study of Clinicians and Observers” (jcm-497382), for publication in Otolaryngology Sectionof Journal of Clinical Medicine. We appreciate the thorough and thoughtful review and for all the insightful comments and suggestions, as the comments have significantly improved our manuscript. We have addressed, on a point-by-point basis, all of the comments made by the reviewers. All changes have been included (marked in red) in our revised submission.
The article was reviewed by a professional English editing company.
Reviewer 1#
1. “First of all, I would like to thank you all for the contribution you have made to the field of craniofacial anatomy. I believe that far too little research effort is placed into understanding normal variation in craniofacial development, and your article helps fill that void. My suggestions for improvement are as follows:”
Answer:We greatly appreciate the feedback from the reviewer as the comments have significantly improved our manuscript.
2. “Introduction-In the first paragraph you talk about fluctuating facial asymmetry and clinically relevant facial asymmetry. I think that it would be useful to distinguish between fluctuating asymmetry and facial asymmetry requiring clinical treatment, which is more commonly directional rather than fluctuating.”
Answer: It was addressed in Introduction section, as requested:
-(called facial asymmetric deformity,which is more commonly directional rather than fluctuating)…
3. “Methods-On lines 97-100 you list the criteria for exclusion, but it was not clear if mixed or uncertain ethnicity needed to be present with craniofacial deformities or if these were two separate exclusion criteria. Logically I would assume that it’s the latter, but if so it would be better to rework this sentence to make it more clear.”
Answer: This sentence was reformulated as requested:
- Individuals were excluded from the study if 1) they had mixed or uncertain ethnicity (non-Taiwanese); 2) had presumed or confirmed diagnosis of any syndromic or nonsyndromic craniofacial deformity; 3) had history of facial trauma; or 4) had undergone orthodontic treatment or facial surgery.
4.Methods-The facial masks in Figure 2 appear to be squished, such that the height is too small for the width of each image
Answer: The Figure 2 was updated as requested.
5. “Methods-In section 2.3 you note that you divided the sample into even groups of 150 based on the severity of asymmetry. While I understand the logic behind this decision based on your panel assessment method, These groups are not inherently meaningful. It would be interesting to know what the distribution of fluctuating asymmetry is in this population. Even a simple dot plot for the entire sample or for the age groups in the sample in your Results section would be very informative.”
Answer: A box plot was included for the levels of asymmetry as requested in question 7. The age groups-related information was also updated as requested in question 6.
6. “Results-The first paragraph in this section lists the age distribution for the sample. This may be a matter of personal preference, but I think this would be more clearly displayed as a table with the actual number of individuals in each age group listed with fluctuating asymmetry data.”
Answer: The actual number of individuals in each age group listed with fluctuating asymmetry data was included in Table 1, as requested.
7. Results-Similarly, section 3.1 lists the fluctuating asymmetry data for each of the severity groups. A box and whisker plot would be a useful way to visualize these data.
Answer: A box plot was included to visualize the four levels of asymmetry as requested:
- Figure 3. Box plot demonstrating the distribution of facial fluctuating asymmetry (n=150 for level of asymmetry). Red asterisks indicate maximum outliers’ values (n=6 for level IV).
8. Conclusions-I would suggest eliminating this section and placing this sentence at the beginning of the Discussion section, unless this section is required by the journal.
Answer: The Journal of Clinical Medicine requests a Conclusion section.
We can provide additional modifications and explanations at the request of the Reviewers and/or Editorial Board. Thank you very much for your consideration of our manuscript.
The author

Reviewer 2 Report
The manuscript describes a comparison of 3D analytic and panel assessment of facial asymmetry in normal pediatric subjects. The manuscript is well organized and well written. I have only a few very minor comments
Line 170: What level of instruction, guidance or training was provided for the observers? I'm not sure how the laypersons would know how to pick a "1" vs "2" vs "4" for asymmetry assessment. Where they given example cases or descriptive text (e.g. "perfectly mirrored" vs "obvious left-right mismatch) to help them judge?
p<0.05: Multiple comparison are being performed here and it is quite likely that the threshold for significant is much lower (e.g. Bonferroni corrections or similar should be applied). I suspect that generally the conclusions of the paper will not change as the significant results were p<0.01, but the authors should examine the validity of "p<0.05" in the multiple tests in Table 2
Line 241: "behind the scope of authors": Not sure what was meant here. Perhaps "beyond the scope of work"?
358: "We did not distribute the facial asymmetry by side or anatomical region, because our objective was to appraise the facial asymmetry overall."
Although a global assessment of asymmetry was performed here, the distribution of local, body feature-specific asymmetries may affect perception of overall asymmetry. (see http://www.ncbi.nlm.nih.gov/pubmed/21355063) For example, asymmetries of the nose may be more perceptible than asymmetries of the chine. If there were more nose asymmetries on one group or another may have skewed results. If the authors should comment on this briefly.
Author Response
May 8, 2019
Prof. Dr. Emmanuel Andrès
Editor-in-Chief,Journal of Clinical Medicine
Dear Prof. Dr. Andrès,
We would like to thank you and your reviewers for considering our manuscript entitled “Identifying Three-Dimensional Facial Fluctuating Asymmetry in Normal Pediatric Individuals: A Panel Assessment Outcome Study of Clinicians and Observers”(jcm-497382), for publication in Otolaryngology Sectionof Journal of Clinical Medicine. We appreciate the thorough and thoughtful review and for all the insightful comments and suggestions, as the comments have significantly improved our manuscript. We have addressed, on a point-by-point basis, all of the comments made by the reviewers. All changes have been included (marked in red) in our revised submission.
The article was reviewed by a professional English editing company.
Reviewer 2#
1. The manuscript describes a comparison of 3D analytic and panel assessment of facial asymmetry in normal pediatric subjects. The manuscript is well organized and well written. I have only a few very minor comments
Answer: We greatly appreciate the feedback from the reviewer as the comments have significantly improved our manuscript.
2. Line 170: What level of instruction, guidance or training was provided for the observers? I'm not sure how the laypersons would know how to pick a "1" vs "2" vs "4" for asymmetry assessment. Where they given example cases or descriptive text (e.g. "perfectly mirrored" vs "obvious left-right mismatch) to help them judge?
Answer: In this study,we adopted a binomial facial symmetry–asymmetry score for appraisal of facial asymmetry in normal pediatric individuals.All raters have received the same level of instruction and guidance before evaluation of 3D image set.The level of instruction and guidance were included in methods section, as requested. We do not have a formal training in this study, with “perfectly mirrored” versus “obvious left-right mismatch” parameters deserving further investigation in pre-adolescent observers. It was also addressed in discussion section:
-Using a spreadsheet per each slide, the rater wrote down his/her perception of facial symmetry versus asymmetry that each individual under appraisal was believed to present. For this, all raters received the same instruction and guidance before appraisal of 3D image set: 1) a description about the purpose of study (how do you perceive the face of this individual?); 2) an elucidation to define the facial symmetry versus asymmetry parameter (one side of the face may be similar or different than the other side, respectively) was provided with each rater paraphrasing it with no comprehension problems; and 3) an explanation about the answering of spreadsheet (mark the circle corresponding to the binominal choice: facial symmetry or facial asymmetry options).
-We adopted a binomial facial symmetry–asymmetry score, and other studies may test additional parameters (perfectly mirrored versus obvious left-right mismatch) andLikert-type or visual analog scales in pre-adolescent observers.
3. “p<0.05: Multiple comparison are being performed here and it is quite likely that the threshold for significant is much lower (e.g. Bonferroni corrections or similar should be applied). I suspect that generally the conclusions of the paper will not change as the significant results were p<0.01, but the authors should examine the validity of "p<0.05" in the multiple tests in Table 2”
Answer: Bonferroni correctionswere adopted formultiple comparisons. Two-sided values of p < 0.05 were considered statistically significant for all analyses, including Table 2. It was addressed as requested:
-Statistical section: Two-sided values of p < 0.05 were considered statistically significant.
-Table 2 (footnote): Adjustment for multiple comparisons (Bonferroni corrections).
4. Line 241: "behind the scope of authors": Not sure what was meant here. Perhaps "beyond the scope of work"?
Answer: “Behind the scope of authors” was replaced by "beyond the scope of work" as requested.
5. 358: "We did not distribute the facial asymmetry by side or anatomical region, because our objective was to appraise the facial asymmetry overall." Although a global assessment of asymmetry was performed here, the distribution of local, body feature-specific asymmetries may affect perception of overall asymmetry. (see http://www.ncbi.nlm.nih.gov/pubmed/21355063) For example, asymmetries of the nose may be more perceptible than asymmetries of the chine. If there were more nose asymmetries on one group or another may have skewed results. If the authors should comment on this briefly.
Answer: We have added this aspect in our manuscript, as requested. This reference is also used in our text as reference 21:
-Although a global assessment of asymmetry was performed here, the distribution of local, body feature-specific asymmetries (e.g., nose versus chin asymmetries) may have affected perception of overall asymmetry [21], deserving further evaluation.
6. The authors describe the ethical review and informed consent provided, but I didn't see an explicit statement that the four individuals (or more precisely their parents) had given consent for use of their likeness in a medical journal (in the supplemental video).
Answer: For this study, all parents of the included individuals provided written consent for participation. Parents of individuals who appear in video 1 provide consent forms which are being submitted to JCM as well as was addressed in the text:
-All parents of the included individuals provided written consent for participation in this study.
-Video S1 – Parents provided written consent for the use of their children images.
We can provide additional modifications and explanations at the request of the Reviewers and/or Editorial Board. Thank you very much for your consideration of our manuscript.
The authors

Reviewer 3 Report
Well written, original, interesting topic.
Author Response
May 8, 2019
Prof. Dr. Emmanuel Andrès
Editor-in-Chief,Journal of Clinical Medicine
Dear Prof. Dr. Andrès,
We would like to thank you and your reviewers for considering our manuscript entitled “Identifying Three-Dimensional Facial Fluctuating Asymmetry in Normal Pediatric Individuals: A Panel Assessment Outcome Study of Clinicians and Observers” (jcm-497382), for publication in Otolaryngology Sectionof Journal of Clinical Medicine. We appreciate the thorough and thoughtful review and for all the insightful comments and suggestions, as the comments have significantly improved our manuscript. We have addressed, on a point-by-point basis, all of the comments made by the reviewers. All changes have been included (marked in red) in our revised submission.
The article was reviewed by a professional English editing company.
Reviewer 3#
1. Well written, original, interesting topic.
Answer:We greatly appreciate the feedback from the reviewer as the comments have significantly improved our manuscript.
We can provide additional modifications and explanations at the request of the Reviewers and/or Editorial Board. Thank you very much for your consideration of our manuscript.
The authors
